# Co-Amorphous Formation of Simvastatin-Ezetimibe: Enhanced Physical Stability, Bioavailability and Cholesterol-Lowering Effects in LDLr−/−Mice

**DOI:** 10.3390/pharmaceutics14061258

**Published:** 2022-06-13

**Authors:** Shamuha Bahetibieke, Sakib M. Moinuddin, Asiya Baiyisaiti, Xiaoang Liu, Jie Zhang, Guomin Liu, Qin Shi, Ankang Peng, Jun Tao, Chang Di, Ting Cai, Rong Qi

**Affiliations:** 1Department of Pharmacology, School of Basic Medical Sciences, Peking University Health Science Center, 38 Xueyuan Road, Haidian District, Beijing 100191, China; 1911210055@bjmu.edu.cn (S.B.); 18299098979@163.com (A.B.); xiaoangliu93@163.com (X.L.); pengak@126.com (A.P.); dichangls@126.com (C.D.); 2School of Pharmacy, China Pharmaceutical University, 24 Tong Jia Xiang, Xuanwu District, Nanjing 210009, China; sa12kib@yahoo.com (S.M.M.); zhangjie448215@163.com (J.Z.); 3320011143@cpu.edu.cn (G.L.); sqzyf1314@163.com (Q.S.); 1721010093@cpu.edu.cn (J.T.); 3College of Pharmacy, California Northstate University, 9700 West Taron Drive, Elk Grove, CA 95757, USA; 4School of Pharmacy, Shihezi University, Shihezi 832000, China; 5Key Laboratory of Molecular Cardiovascular Sciences, Ministry of Education, 38 Xueyuan Road, Haidian District, Beijing 100191, China; 6NHC Key Laboratory of Cardiovascular Molecular Biology and Regulatory Peptides, 38 Xueyuan Road, Haidian District, Beijing 100191, China; 7Beijing Key Laboratory of Molecular Pharmaceutics and New Drug Delivery Systems, 38 Xueyuan Road, Haidian District, Beijing 100191, China

**Keywords:** hypercholesterolemia, co-amorphous, simvastatin, ezetimibe, cholesterol-lowering drug

## Abstract

Hypercholesterolemia is one of the independent risk factors for the development of cardiovascular diseases such as atherosclerosis. The treatment of hypercholesterolemia is of great significance to reduce clinical cardiovascular events and patient mortality. Simvastatin (SIM) and ezetimibe (EZE) are commonly used clinically as cholesterol-lowering drugs; however, their treatment efficacy is severely affected by their poor water solubility and low bioavailability. In this study, SIM and EZE were made into a co-amorphous system to improve their dissolution, oral bioavailability, storage stability, and cholesterol-lowering effects. The SIM-EZE co-amorphous solids (CO) were prepared successfully using the melt-quenched technique, and the physicochemical properties of CO were characterized accordingly, which exhibited improved physical stability and faster dissolution release profiles than their physical mixture (PM). In the pharmacokinetic study, the SIM-EZE CO or PM was given once by oral gavage, and mouse blood samples were collected retro-orbitally at multiple time points to determine the plasma drug concentrations. In the pharmacodynamic study, low-density lipoprotein receptor-deficient (LDLr−/−) mice were fed with a high-fat diet (HFD) for two weeks to establish a mouse model of hypercholesterolemia. Using PM as a control, we investigated the regulation of CO on plasma lipid levels in mice. Furthermore, the mice feces were collected to determine the cholesterol contents. Besides, the effect of EZE on the NPC1L1 mRNA expression level in the mouse intestines was also investigated. The pharmacokinetics results showed that the SIM-EZE CO has improved bioavailability compared to the PM. The pharmacodynamic studies showed that SIM-EZE CO significantly increased the cholesterol-lowering effects of the drugs compared to their PM. The total cholesterol excretion in the mouse feces and inhibitory effect on NCP1L1 gene expression in the mouse intestines after being given the SIM-EZE CO were more dramatic than the PM. Our study shows that the SIM-EZE CO prepared by the melt-quenched method can significantly improve the stability, bioavailability, and cholesterol-lowering efficacy with excellent development potential as a new drug formulation.

## 1. Introduction

Hypercholesterolemia (HC) is usually caused by diet, metabolic diseases, or inherited diseases [1]. The prevalence of HC is increasing with the improvement of people’s living conditions. It is believed that the prevalence of HC in most ethnic groups throughout the world is about 1:500 [2,3]. According to the Chinese guidelines on prevention and treatment of dyslipidemia (2018 version), individuals with total cholesterol (TC) greater than 5.7 mmol/L can be diagnosed as HC [4]. A disorder of TC level can affect the metabolism of low-density lipoprotein (LDL), causing a large amount of LDL cholesterol accumulation in the body and eventually leading to various cardiovascular diseases (CVD) at a very young age [5]. At present, dietary regulation [6,7] and lifestyle modification [8,9] are the primary treatments in clinical practice. Based on this, statins are commonly used to reduce the cholesterol level in treating patients with simple hypercholesterolemia [10]. It was reported that the application of statins could reduce the relative risk of coronary heart disease events by 30–35% [11]. In addition to significantly reducing total cholesterol and LDL cholesterol, statins can also slightly increase high-density lipoprotein (HDL) cholesterol and lower triglyceride (TG) levels in plasma, thus directly reducing the mortality of patients with cardiovascular diseases [12,13]. Beyond that, drugs such as fibrates, nicotinic acid, and cholestyramine are used as adjunctive therapy in clinics to treat hyperlipemia according to the different conditions of patients [14].

Simvastatin (SIM) (Figure 1) dampens endogenous synthesis of cholesterol by inhibiting the activity of 3-hydroxy-3-methyl glutaryl coenzyme A (HMG CoA) reductase, a rate-limiting enzyme of cholesterol biosynthesis [15], thus reducing the plasma cholesterol level [16,17]. SIM is a highly permeable drug with a solubility of 1.45 mg/L, and its oral bioavailability is only about 5% [18]. Ezetimibe (EZE) (Figure 1) is a cholesterol-lowering drug widely used in clinical practice [13,19]. EZE mitigates the exogenous absorption of sterol and phytosterol by selectively blocking Niemann-Pick C1-Like 1 (NPC1L1) protein, thus inhibiting cholesterol transport from the small intestine to the liver and leading to reduced liver cholesterol storage. Besides, EZE increases liver LDL receptor synthesis to accelerate LDL metabolism [20], thus decreasing plasma cholesterol levels. EZE has a saturated solubility of 8.46 mg/L and approximately 35% oral bioavailability [21]. According to the Biopharmaceutical Classification System (BCS), SIM and EZE have been classified as Class II drugs with poor water solubility but good permeability [22].

Co-amorphous (CO) is a single-phase amorphous binary system, comprised of active pharmaceutical ingredients (APIs) and other physiologically acceptable small molecular excipients through non-covalent bonds (such as hydrogen bonds), ionic bonds, or non-bonding interaction [15,16,23,24]. CO can improve the stability of amorphous monomers and enhance the solubility and dissolution rates of drugs [16,24,25]. In recent years, the combined use of drugs has received extensive attention in clinical practice. According to the concept of synchronous release, when two drugs form a CO system, they can treat two different diseases or synergistically treat the same disease, thereby improving clinical efficacy and reducing side effects [16,21,24,26].

In clinical practice, ezetimibe (EZE) and simvastatin (SIM) are commonly used to decrease cholesterol, but their therapeutic effectiveness is significantly hampered by their poor water solubility and limited bioavailability. In this study, we created a CO system of SIM and EZE based on the different mechanisms of their cholesterol-lowering effects and the defects in the dissolution and absorption of these two drugs. The formation of CO can stabilize the high-energy binary amorphous mixture, make it easier for drug monomers to be absorbed, and reduce the levels of cholesterol in mouse plasma.

## 2. Methods

### 2.1. Materials

Simvastatin (SIM) and Ezetimibe (EZE) were purchased from Jiangsu Zenji Pharmaceuticals Co., Ltd. China (purity > 99.0%) Acetonitrile (ACN) and phosphoric acid (H_3_PO_4_) were purchased from Nanjing Chemical Reagent Co., Ltd. (Nanjing, China). Triethylamine was purchased from Sinopharm Chemical Reagent Co., Ltd. (China). Other reagents in HPLC grade were purchased from Alligator Reagent (China). Cholesterol Assay Kit and Triglyceride Assay Kit were purchased from Beijing Zhongsheng Beikong Biotechnology Company. The simulated solutions of gastric fluid (pH 1.6) and intestinal fluid (pH 6.5) were prepared according to the pharmacopoeia methods [27].

### 2.2. Preparation of SIM-EZE PM and SIM-EZE CO

Initially, a vortex mixer was used to prepare the PM of SIM and EZE (1:1 molar ratio). In this study, 1 g of SIM-EZE PM was placed in an aluminum block. Then the sample was transferred to an oven and allowed to heat until it melted totally, and the melted sample was kept for 5 min. At this stage, liquid nitrogen was placed in a stainless-steel tray. The melted sample was immediately transferred to the top of the tray, and a few drops of liquid nitrogen were poured on the top of the sample. This melt-quenched CO sample was then transferred to the desiccator at room temperature. For testing the physical stability, the individual amorphous drugs were prepared by the melt-quenched method for comparison.

### 2.3. Characterization of CO

#### 2.3.1. Thermogravimetric Analysis (TGA)

The TGA analysis was performed by Q500 (TA instrument, New Castle, DE, USA) under a dynamic nitrogen atmosphere with a flow rate of 50 mL/min. Then the prepared sample powders were heated to the desired temperature at a heating rate of 20 °C/min.

#### 2.3.2. Differential Scanning Calorimetry (DSC)

The sample thermal transitions were measured by the Q2000 DSC instrument (TA instrument, New Castle, DE, USA). The *T_g_* was determined as the midpoint of the change of heat capacity, and crystallization temperature (*T_c_*) and melting point temperature (*T_m_*) were measured as the onset temperatures.

#### 2.3.3. Fourier Transform Infrared Spectroscopy (FT-IR)

The molecular interaction between SIM and EZE was investigated by FT-IR spectroscopy (Thermo Fisher, Nicolet IS10). The spectra were recorded over a range of 4000–400 cm^−1^ with a resolution of 4 cm^−1^.

#### 2.3.4. Physical Stability

For testing the physical stability, pure amorphous drugs and CO were stored with desiccators at 4 °C and 40 °C. The X-ray powder diffraction (XRPD) was used to investigate the recrystallization behavior of fresh, 7, 14, and 30 day samples. The XRPD patterns were collected by a D8 Advance X-ray Diffractometer (Bruker, Germany) using Cu Kα radiation (λ = 1.5406 Å) at 40 kV and 40 mA. All diffraction data were collected from 3 to 40° (2θ) with a step size of 0.02° and a step time of 1 s.

#### 2.3.5. In Vitro Powder Dissolution

Powder dissolution test under sink condition was conducted by a dissolution tester obtained from Tianjin Xinzhou Science & Technology Co., Ltd (Tianjin, China). Under-sink condition, 1:1 molar ratio of SIM-EZE CO and their PM were taken in 500 mL of 60% ethanol- 40% purified water (*v*/*v*) as dissolution media with a paddle speed 50 rpm at 37 °C. A 200 mg of each sample was taken into the dissolution media, and the samples were collected at 5, 10, 20, 30, 45, and 60 min. Notably, 2 mL aliquots were withdrawn and replaced with a 2 mL fresh dissolution medium at 37 °C. A 0.22 µm nylon disk filter filtered all samples, and then the drug release was determined by the high-performance liquid chromatography (HPLC).

#### 2.3.6. HPLC

All dissolution test samples were analyzed by a prominence HPLC system of Shimadzu, containing a UV detector (SPD 20A) and equipped with a C18 column (250 × 4.6 mm) purchased from Agilent. The HPLC mobile phase consisted of 65% acetonitrile and 35% ammonium acetate (pH 5). The column oven temperature was kept at 30 °C. The sample injection volume was 20 µL, and the flow rate was 1.2 mL/min. The detection wavelength was fixed at 240 nm, the retention time of EZE and SIM was 3.27 min and 11.34 min, respectively. An excellent linear curve (r = 0.999) was obtained over the drug concentration range of 40–400 µg/mL.

#### 2.3.7. Gordon–Taylor Equation for Theoretical *T_g_* Calculation

The theoretical *T_g_* of an amorphous binary system can be predicted by the Gordon–Taylor equation as follows
(1)$$T_{g\left(mix\right)}=\frac{w_{1}\cdot T_{g1}+K\cdot w_{2}\cdot T_{g2}}{w_{1}+K\cdot w_{2}},\;$$
where *T_g(mix)_* is the theoretical *T_g_* of CO, and *T_g_*_1_ and *T_g_*_2_ are the glass transition temperature of each drug, *w*_1_ and *w*_2_ are weight fractions, and *K* is a constant, which can be further expressed as:(2)$$K=\frac{T_{g1}\cdot \rho_{1}}{T_{g2}\cdot \rho_{2}},\;$$
where $\rho_{1}$ and $\rho_{2}$ are the respective powder densities of EZE and SIM. The powder density of EZE and SIM was 1.34 g/m^3^ [28] and 1.11 g/m^3^ [29], respectively.

### 2.4. In Vivo Pharmacokinetic Study

C57BL/6 WT mice were raised in SPF animal house in the animal breeding center of Peking University Health Science Center, and temperature and relative humidity were stabilized at 24 °C and 45%, respectively. The experimental mice’s feeding and application were in accordance with the National Institutes of Health Laboratory Animal Application Guide and approved by the Animal Ethics Committee of the Peking University School of Healthcare (Ethical Code: LA2018270; Date: 31 August 2018).

In the pharmacokinetics study, the 16 WT mice were randomly and equally allocated into 2 groups and treated with different SIM-EZE formulations, namely CO group (described as CO) and PM group (described as PM). All mice fasted overnight. The CO or PM of SIM-EZE has been suspended in 0.5% carboxymethyl cellulose sodium (*w*/*v*) aqueous solution and given to the mice at a single dose of 5 mg/kg by gavage. The blood samples were collected retro-orbitally at 0.5, 1, 1.5, 2, 3, 4, 6, 8, 10, 12, 14, and 24 h, respectively. The plasma samples were treated in the following order to measure the concentrations of SIM and EZE in the blood: (1) 50 μL plasma and 20 μL SIM or EZE standard solution were dissolved in methanol and vortexed for 10 s; (2) 150 μL dichloromethane and 450 μL cyclohexane were added and vortexed for 3 min; (3) the sample was centrifuged at 4000 rpm, 4 °C for 15 min, and the supernatant was collected; (4) the supernatant was placed in a water bath at 40 °C to evaporate the solvent by purging it with nitrogen gas; (5) the resulting residues were reconstituted with a 200 μL mobile phase and vortexed for 3 min; (6) the resultant sample was centrifuged at 1000 rpm, 4 °C for 15 min, and the supernatant was collected for measurement [30].

All the samples were separated under isocratic elutions using acetonitrile (A): 0.02 mol/L sodium dihydrogen phosphate solution (65% A) for 20 min and a wavelength of 238 nm was used for the detection of the plasma concentration of SIM (the plasma EZE concentrations were too low to be detected by HPLC in the experimental conditions).

### 2.5. Pharmacodynamic Study on the Hypolipidemic Effects of the CO

Thirty-two LDLr−/− male mice, weighing 20–25 g and aged 7–8 weeks, were purchased from the animal breeding center of Peking University Health Science Center (Beijing, China); all mice were raised under a 12-h light/dark cycle with free access to food and water. Among them, 12 mice were used in SIM-EZE PM experiments to select an appropriate dose (2.5, 5, or 10 mg/kg/day) for the subsequent hypolipidemic experiments of the CO, and the remaining 15 mice were used for the pharmacodynamic evaluation of SIM-EZE CO in the treatment of hypercholesterolemia.

For pharmacodynamic studies, the 15 mice were allocated into 3 groups randomly and equally, namely model group (model), CO group (CO), and PM group (PM). First, the mice were fed with a high-fat diet (HFD) made of 0.5% cholesterol and 20% lard. Then the mice in the model group were given a 0.5% CMC-Na solution. The mice in the other two groups were given a 5 mg/kg/day CO or PM of SIM-EZE. The drugs were dissolved in 1 mL 0.5% CMC-Na solution and given orally by gavage daily for two weeks and along with HFD feeding. The mice were fasted overnight and sacrificed two weeks later for sampling.

### 2.6. Plasma TC and TG Measurement

In the pharmacodynamic study, the mice were anesthetized according to the guidelines of National Institutes of Health (NIH). The blood samples were collected from the orbital venous plexus of mice and put into centrifuge tubes pretreated with heparin. Then the samples were centrifuged at 826 g (5810 R high-speed desktop centrifuges, Eppendorf, Hamburg, Germany), 4 °C for 10 min, and the plasma was collected for the determination of TC and TG levels by the kits (BioSino Bio-Technology and Science Inc., Beijing, China). The plasma TC and TG levels were monitored before the experiment and after 1 week and 2 weeks of drug administration.

### 2.7. Determination of Cholesterol in the Mouse Feces

The purpose of this study was to investigate the effects of EZE on the cholesterol excretion in mice feces; a total of 12 mice were divided into 4 groups on average: control group, model group, CO group, and PM group. The mice were fed with HFD for one week except those in the control group (fed with a chow diet). After feeding the HFD for one week, the mice in the drug treatment group were given 5 mg/kg/day CO or PM of SIM-EZE by gavage for 3 days and maintained HFD feeding. After that, the mice had fasted overnight, and the feces produced by the mice within 24 h were collected and treated using the following method. First, the feces were dried in an oven for 24 h and ground after that. An organic solvent (chloroform: methyl alcohol = 7:3) with a volume of 3 mL was added into 0.3 g ground feces. The sample was placed in a water bath at 45 °C for 24 h and then centrifuged at 3000 rpm for 10 min. Next, the supernatant was taken into a new EP tube, and 0.5 mL of normal saline was added and vortexed for 30 s to remove the soluble impurities. Then the sample was centrifuged at 3000 rpm for 10 min. Finally, the lower layer of organic solution was taken to determine the cholesterol level by the kits.

### 2.8. RNA Extraction and q-PCR Analysis

The mouse intestinal tissues were collected from the Experimental Section 2.7. According to the manufacturer’s instructions, total RNA was extracted from the mouse intestinal tissues using Trizol reagent, and mRNA was reverse transcribed into cDNA using RT MasterMix. Then the q-PCR analysis was performed using EvaGreen qPCR MasterMix in MiniOpticon real-time PCR detection system.

### 2.9. Statistical Analysis

The results of statistical analysis were represented as mean ± standard deviation (Mean ± SEM). The data were statistically analyzed using GraphPad Prism 5.01 software (GraphPad Prism Inc., San Diego, CA, USA) and the One-Way ANOVA test, and *p* < 0.05 was considered statistically significant.

## 3. Results

### 3.1. Characterization of SIM-EZE CO

The SIM-EZE CO was prepared using the melt-quenched method, by which the PM was heated until it melted and then immediately quenched at low temperatures. The key to this approach was that the melted sample was cooled down from the high temperature, so the disordered drug molecules had no time to rearrange their molecular orientations to recrystallize.

#### 3.1.1. Thermal Analysis

In order to test the thermal stability of drugs for preparing CO mixtures by the melt-quenched method, the TGA of crystalline SIM and EZE is shown in Figure 1. Both drugs exhibit good thermal stability. It was worth noting that EZE was starting to have a slight weight loss at 45 to 90 °C. It has been reported that the mass loss of hydrate EZE below 100 °C was due to the loss of water molecules in the EZE [31,32,33]. SIM was thermally stable below its melting point from the TGA trace (Figure 1).

Figure 2 presents the DSC traces of crystalline EZE, SIM, and their 1:1 molar ratio of PM. Crystalline EZE has shown two melting peaks, where the first onset endothermic peak was observed at 73.69 °C and the second one at 163 °C. It was reported that hydrate EZE loses water at 40 to 70 °C [33], We observed an endothermic dehydration peak at 73.69 °C and then the melting peak of crystalline EZE at 163 °C (Figure 2A). Crystalline SIM showed a characteristic melting peak at 138.75 °C. As shown in Figure 2A, the crystalline PM started to melt at the temperature much lower than the melting points of two drugs, and a eutectic melting point at 117.89 °C was observed for the PM of EZE and SIM at 1:1 molar ratio.

Amorphous samples had a characteristic *T_g_*, followed by no endothermic peak, confirming they were in the amorphous state (Figure 2B). The amorphous EZE and SIM showed a clear *T_g_* at 65.2 °C, and 34.5 °C, respectively, and their 1:1 molar ratio of co-amorphous showed a single *T_g_* at 53.45 °C, which was near to the middle of the *T_g_*s for two amorphous drugs. Notably, Knapik-Kowalczuk et al. reported that SIM-EZM midpoint *T_g_* was 50 °C, which was similar to our experimental result [19].

The Gordon–Taylor equation (Equation (1)) has been widely employed to study the glass transition temperature of amorphous binary systems. In this study, the predicted *T_g_* of SIM-EZE CO was calculated by Equations (1) and (2). The predicted *T_g_* was 47.52 °C, which was 5.93 °C lower than the experimental *T_g_* of the CO (53.45 °C). The deviation between the experimental and the predicted *T_g_* could typically indicate the presence of molecular interactions between the molecules of multicomponents [34]. Therefore, the positive deviation of *T_g_* indicated the formation of molecular interactions between SIM and EZE in the CO.

#### 3.1.2. FT-IR Spectroscopy

From the literature, a CO system could be formed by molecular interaction, salt formation, anti-plasticizing effects, and intimate mixing [15,23,35]. Among them, molecular interaction via hydrogen bonding was most commonly reported, and this interaction can be responsible for enhancing physical stability and improving their dissolution rates [15,16,23,24]. Our previous studies pointed out that atenolol-hydrochlorothiazide and febuxostat-indomethacin CO systems formed via hydrogen bonding interactions between the heterodimers [16,24]. In this work, FT-IR was used to characterize the molecular interactions of CO by comparing with crystalline and amorphous forms of the individual components (SIM and EZE), PM, and CO sample.

As shown in Figure 3A, a sharp peak was noticed at 1718.3 cm^−1^, attributed to C=O stretching of EZE; a similar observation was reported by Jahangiri et al. [36]. Riekes et al. reported that amorphous EZE shifted from 1725 to 1717 cm^−1^ region [28]; however, Alhayali observed that the C=O band of amorphous EZE was moved from 1728 to 1723 cm^−1^ [32]. Our experimental result is similar to their results, i.e., C=O stretching of amorphous was slightly blue-shifted from 1718.3 to 1722.3 cm^−1^ (Figure 3A). In addition, the -OH stretching peaks of crystalline EZE above 3000 cm^−1^ broadened in amorphous EZE [32]. After the amorphization, the disordering of crystal lattice structure and solid-state molecular rearrangement could cause peak broadening [16,24].

In Figure 3B, the crystalline SIM exhibited a stretching vibration in the 1701.78 cm^−1^ regions, while Laitinen et al. reported that the crystalline SIM had a stretching vibration at 1695 cm^−1^ region and amorphous SIM shifted from 1695 to 1714 cm^−1^ region [37]. In addition, an O-H stretching of SIM was observed at the 3550.3 cm^−1^ region. Notably, this peak shifted and broadened from 3550.3 cm^−1^ to 3459.5 cm^−1^ in amorphous powder (Figure 3B).

In the SIM-EZE CO sample, the C=O stretching was shifted from 1714 to 1718 cm^−1^, as well as broadened the -OH stretching from the PM (Figure 4). Notably, significant peak broadenings were observed in atenolol-hydrochlorothiazide, febuxostat-indomethacin, and ritonavir-indomethacin co-amorphous solids with the presence of molecular interactions [16,24,38]. Our findings suggested that hydrogen bonds were formed between the C=O stretching and -OH stretching in the SIM-EZE CO. In addition, the Gordon–Taylor equation showed a positive deviation from the theoretical *T_g_*, which would indicate molecular interactions between each component of the mixture.

#### 3.1.3. Physical Stability

In order to test the physical stability, the individual amorphous drugs and CO were stored at 4 °C and 40 °C. As shown in Figure 5, the stability of SIM-EZE CO was compared with the neat amorphous drugs of SIM and EZE. The samples were evaluated at 0, 7, 14, and 30 days by XRPD, and the results indicate that SIM-EZE CO has better physical stability than the individual amorphous components. In this study, all samples were stable at 4 °C. However, at 40 °C, both amorphous EZE and SIM were partially recrystallized during 30 days of stability test. In contrast, the CO sample was quite stable during the stability experiment, without showing any diffraction peaks in the XRPD patterns after 30 days.

#### 3.1.4. In Vitro Powder Dissolution

In Figure 6, a fast dissolution profile was observed for EZE and SIM in the CO sample than their PM. The EZE in CO improved 1.50, 1.26, and 1.01 folds compared to EZE in the PM at 5, 10, and 20 min, respectively. After 30 min of dissolution, both of the CO and PM reached 100% release. The SIM in CO improved 1.41 and 1.13 folds compared to SIM in the PM at 5 and 10 min, respectively. However, after 20 min, the SIM in the CO and PM had a similar release pattern.

### 3.2. Pharmacokinetic Study

Both SIM and EZE have low solubility and bioavailability, so we investigated whether the SIM-EZE CO could improve the bioavailability of the two drugs. We conducted pharmacokinetic studies on WT male mice and then performed a comparative analysis between CO and PM. Since EZE only has 10–20% prototype in the blood after oral administration, and its bioavailability is unmeasurable [39], we determined the plasma concentration-time curve of SIM. We used software DAS 2.0 (Mathematical Pharmacology Professional Committee of China, Shanghai, China) to fit the pharmacokinetic data of mice, and the results showed that the one-compartment model was most suitable and had the best fitting degree. As shown in Figure 7, the *T*_max_ of the two systems reached 2 h. The CO group had the highest *C*_max_ compared to the PM group. The results of pharmacokinetic parameter analysis in Table 1 also showed that the SIM-EZE CO prepared by the melt-quenched method had improved bioavailability compared to the PM.

### 3.3. Dose-Dependent Cholesterol-Lowering Effects of the SIM-EZE PM

The mice were given different doses of PM to investigate their effects on lowering plasma cholesterol. A dose of 10 mg/kg/day of SIM and EZE is commonly used in clinical practice to treat patients with hyperlipidemia [40]. In order to investigate whether a co-amorphous system could maximize the efficacy of the drug using a small dose, the PM was given by gavage at the doses of 2.5, 5, or 10 mg/kg/day for 21 days, and the mice were fed with HFD at the same time. The mouse plasma cholesterol levels were tested at 0, 7, 14, and 21 days. The results showed in Figure 8 that a 10 mg/kg/day PM could significantly reduce the plasma cholesterol level of mice on the 7th day than the HFD-fed mice but not given drugs. The plasma cholesterol level did not reduce until the 21st day when a 5 mg/kg/day PM was administrated to the mice. The dose of 5 mg/kg/day was finally selected to investigate whether a co-amorphous system can improve drug release and dissolution.

### 3.4. Pharmacodynamics Studies of the SIM-EZE CO and PM

The binary PM and the SIM-EZE CO were given by gavage to the mice with a dose of 5 mg/kg/day for two weeks, and the mice were fed with HFD during the drug treatment. The changes of the body weight were monitored every day for 14 days, and mouse plasma cholesterol and triglyceride levels were detected at 0, 7, and 14 days, and the results were compared among the HFD model group (model), PM group, and CO group at the same dose of 5 mg/kg/day. Figure 9A showed that the mice body weight did not change significantly after administering the PM or CO sample, indicating the safety of this particular drug dose in their systems. However, the plasma cholesterol levels of LDLr−/− mice increased significantly after being fed with HFD (Figure 9B,C), indicating the successful establishment of the hypercholesteremia mouse model. After 7 days of administering the two-drug formulations, the SIM-EZE CO group reduced more plasma cholesterol and triglyceride levels than the PM group. After 14 days of drug administration, both systems could significantly decrease the mouse plasma cholesterol level, but the CO group was more efficient than the PM group.

### 3.5. Determination of Cholesterol Content in Mouse Feces and Effects of EZE on the mRNA Level of NPC1L1 in the Mouse Intestines

In order to study the effect of EZE on the excretion of cholesterol, the mice were administrated in two systems by gavage while feeding with HFD, and their feces were collected to determine the content of cholesterol [41,42]. As shown in Figure 10A, the results showed that the mice’s cholesterol excretion in the model group was higher compared to the control group. The total cholesterol excretion given in the CO group was lower than that of the mice given the PM.

We investigated the effect of EZE on the NPC1L1 mRNA expression level of mice. As shown in Figure 10B, the results showed that compared with the control group, the NPC1L1 mRNA expression level of mice in the model group was significantly increased. While compared with the model group, the NPC1L1 mRNA expression levels of mice in the CO and PM groups significantly decreased. However, the CO group was the most efficient, indicating that EZE in the co-amorphous could dramatically down-regulate the expression of NPC1L1 and inhibit cholesterol transport and absorption.

## 4. Discussion

CO is not a simple PM of amorphous monomers, but rather a unified system formed by mixing the components at molecular level. EZE and SIM are used to prepare CO drugs because EZE inhibits the exogenous absorption of cholesterol metabolism, while SIM inhibits the endogenous synthesis of cholesterol. These two drugs may have a synergistic effect when they are used together for lowering plasma cholesterol levels from different mechanisms. In addition, EZE does not affect the concentration of statins, and the combination of the two drugs will not cause the compatibility of the drugs to affect the efficacy. Since both drugs are poorly water-soluble and poorly absorbed in the body, the formation of a CO system can increase the dissolution rates of two drugs and maintain them in the amorphous state during storage.

In this study, the SIM-EZE CO was prepared by the melt-quenched method, where the PM was heated to a molten state and then quenched into a solid at a low temperature either by liquid nitrogen to obtain the CO. The CO was then stored in the desiccator at room temperature for an hour. The molten sample was cooled rapidly, which made it impossible for the disordered drug molecules to rearrange and crystallize. Ultimately the crystalline solid was converted into the amorphous form. This method is simple, economical, green technology, and does not involve solvents in the preparation. However, only thermally stable drugs are suitable for this method.

Many studies have reported that the CO powders often have a higher dissolution and a better pharmacokinetics profile than their PM and individual drugs [15,23]. In this work, we used ethanolic solution as a dissolution media. We select ethanolic solution because both drugs suffer poor solubility problems; therefore, it is challenging to perform the dissolution in the pure aqueous media. Gunnam et al. and Goud et al. used a 40% ethanolic solution as dissolution medium to dissolve curcumin cocrystal, CO, etc. [43,44]. We performed a comparative dissolution experiment between the CO and PM samples. The dissolution release rate of SIM and EZE in the CO was faster than SIM and EZE in the PM. In the initial 20 min we saw a rapid dissolution rate in the CO sample, and after 30 min both systems showed 100% release (Figure 6). These faster in vitro CO powder dissolution and release profiles were consistent with our in vivo pharmacokinetics study for SIM.

While fed with an HFD to create a hypercholesterolemia model, mice were continuously given the SIM-EZE CO by oral gavage, with the SIM-EZE PM as a control, in order to investigate whether a CO system can improve the drug lipid-lowering efficacy. Notably, after 7 days of treatment, the SIM-EZE CO could significantly reduce the cholesterol and triglyceride levels in mice plasma, which was consistent with the results of pharmacokinetic studies, in which the CO had the highest *C*_max_ and bioavailability than the PM. Besides, clinical studies reported that both SIM and EZE could reduce plasma TG levels by 5–15% [45]; therefore, we also observed the most efficient TG-lowering effect of CO, except for its TC-lowering-effect.

The literature shows that EZE is rapidly absorbed in the small intestine after oral administration and glycosylated in the liver [39]. After entering the blood, the prototype EZE is only 10–20%, and its active metabolite of glucoside is about 80–90% [46]. So, it has been reported that the bioavailability of EZE is unmeasurable [39]. In this study, both HPLC and LC-MS were unable to determine the plasma concentration of EZE in the mice. Therefore, other physiological and biochemical indicators were used to indirectly prove its pharmacodynamics effects. Since the pharmacological effect of EZE is to block cholesterol absorption from the intestine into the blood, we tested the amount of cholesterol excretion in mice feces to study whether CO could promote the efficacy of EZE. The results showed that the cholesterol excretion of the two-drug administration groups was higher than that of the model group, indicating that EZE can significantly inhibit intestinal cholesterol absorption. Interestingly, the cholesterol excretion in the PM group is the highest in the two-drug treating groups. This could be attributed to the fact that the HFD increased the solubility of EZE in the intestines of mice, leading to the enhanced absorption. It is well known that lipids in the HFD and intestinal bile acids will make poorly soluble drugs solubilized into micelles, thereby increasing their dissolution and absorption [47].

Based on this, we investigated the effect of EZE on the NPC1L1 mRNA expression level of mice (Figure 10). The results showed that compared with the control group mice, the NPC1L1 mRNA expression level in the model group was significantly increased, indicating that the plasma TC level elevated by HFD feeding is correlated with its up-regulation in NPC1L1 expression levels and consequent enhancement in TC transport. Compared to the model group, the NPC1L1 mRNA expression levels of mice in the two-drug administration groups were significantly decreased, indicating that EZE treatment significantly regulated the expression of NPC1L1, thereby selectively inhibiting the absorption of cholesterol, reducing the cholesterol level in plasma. While the CO has more intense inhibition in NPC1L1 expression than the PM, it also explained that EZE in the CO was more efficient in promoting cholesterol efflux from brush border membrane-to-lumen of the intestinal epithelia and excretion of cholesterol out of the mouse intestines in their feces.

However, PMs have sound effects on promoting cholesterol excretion and inhibiting NPC1L1 expression, but the impact of reducing HFD-induced hyperlipidemia in the body is not as good as CO drugs. It may be because the endogenous synthesis of cholesterol has a more significant impact on blood lipid levels than exogenous food intake [48], and the CO significantly improves the absorption of SIM. Therefore, the CO inhibits cholesterol synthase more effectively and produces a stronger effect of lowering blood lipids.

## 5. Conclusions

In summary, the SIM-EZE co-amorphous solid was successfully prepared by the melt-quenched method. The SIM-EZE CO was physically stable at elevated temperature in comparison with the individual amorphous components. A faster dissolution rate, enhanced oral bioavailability, and lipid-lowering effects were observed for the CO sample than the PM. The FT-IR analysis and positive deviation in the predicted *T_g_* indicated a strong molecular interaction between SIM and EZE in the CO system. Furthermore, it was found that SIM-EZE CO exhibit a better effect on decreasing plasma cholesterol levels in LDLr−/− mice fed with HFD than those of their PM. The pharmacokinetic study also supported the conclusion that the two drugs in the CO had higher absorption than the PM. Thus, we believe that the SIM-EZE CO may become a potential clinical formulation of combination drugs for the treatment of hypercholesterolemia.

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
