# Peer review of "Co-Amorphous Formation of Simvastatin-Ezetimibe: Enhanced Physical Stability, Bioavailability and Cholesterol-Lowering Effects in LDLr−/−Mice"

_pharmaceutics, 2022, doi:10.3390/pharmaceutics14061258_

Round 1

Reviewer 1 Report

This manuscript presents the  formation of a coamorphous system of two drugs designed to lower blood cholesterol.  The paper is very sound scientifically with proven clinical activity.  I would suggest that the paper is considered for publication following clarification of the following points

(i)  The system described in this paper has been presented previously and therefore the authors must present evidence of the impact of the paper (given that  much of the paper concerns the in vitro characterisation of the drug system)

(ii)  What is the state of the drug mixture following dispersion into the aqueous vehicle and what is the effect of NaCMC on this state

(iii)  I am unclear of the benefits of this new CO mixture.  It would seem that the CMax of the CO and the PM are different but there is no variability around the Cmax of the CO. This is most irregular.  The variation of the Cmax for the PM is much greater than 0.371 ug/mL (according to the graph). Please correct this.  Statistically there is no difference between the Tmax, half life and possibly AUC.  I am unconvinced by the statistical analysis of the data.

Why is the variation so small in the clinical studies?  This is not the normal case

Author Response

Response to Review 1 Comments

(i)  The system described in this paper has been presented previously and therefore the authors must present evidence of the impact of the paper (given that much of the paper concerns the in vitro characterisation of the drug system)

Answer: Previously published paper mainly studied the physical characterization and liquid dynamics of SIM-EZE coamorphous system. However, this manuscript focused on the enhanced physical stability, bioavailability, and cholesterol-lowering effects in vivo for this coamorphous system. The impact of this manuscript is the pharmacodynamics of coamorphous pharmaceuticals, which has rarely been reported in the literature.

(ii)  What is the state of the drug mixture following dispersion into the aqueous vehicle and what is the effect of NaCMC on this state.

Answer: CMC-Na is used as a suspending/thickening agent. We used it to suspend drug for uniform dose adjustment during oral gavage preparation.

(iii)  I am unclear of the benefits of this new CO mixture. It would seem that the CMax of the CO and the PM are different but there is no variability around the Cmax of the CO. This is most irregular.  The variation of the Cmax for the PM is much greater than 0.371 ug/mL (according to the graph). Please correct this.  Statistically there is no difference between the Tmax, half life and possibly AUC.  I am unconvinced by the statistical analysis of the data.

Answer: In general, the half-life (t1/2) is related to the nature of the drug, not the physical change. Therefore, the t1/2 of the same drug, like SIM, should be the same between the CO and the PM.

The in vitro drug release experiment showed that the dissolution rate of SIM from CO was higher than that of PM, and the Tmax of CO should be lower than that of PM theoretically. However, there was no difference between CO and PM at the existing blood collection point-in-time, which was set not to be too densely for the reason of the health status of mice. Therefore, the blood drug concentration curve only could be described roughly. On the other hand, we used oral administration, and the complex environment of gastrointestinal tract may affect the stability of CO, leading to the precipitation of part of SIM, thus reducing the difference in Tmax and AUC between the CO and the PM. In subsequent studies, we will consider intravenous administration or coating the CO to reduce the interference of gastrointestinal tract. In conclusion, CO is still better than PM in such a complex situation in terms of its pharmacokinetic and pharmacodynamic results, so we believe it will have better application prospects in the future.

Why is the variation so small in the clinical studies?  This is not the normal case

Answer: Does the clinical study mentioned by the reviewer in the comments refers to our in vivo studies in mice? All the variations in our experimental data are in the normal range.

Reviewer 2 Report

The manuscript presents a new formulation based on coamorphous formation to administrate two drugs together, Simvastatin and Ezetimibe. Both drugs are administrated in hypercholesterolemia disease, but their therapeutic action are different. Although this type of formulation is already used by some of the authors with other drugs, the application to these drugs can be interested because their synergic effect.

In general, the characterization of the formulation is well reported, and the cholesterol-lowering effect in mice is compared with the administration of single drug and/or physical mixture. In my opinion this work can be publish in this Journal, but before some changes would have to be made.

One of my objections is that the manuscript is too long, and some aspects or comments are repeated in several sections. The authors should try to reduce the text without losing its integrity. The Abstract should be rewriting as the Instructions for Authors recommendations.

Some other aspects to be considered point by point:

The structures of the compounds should be reported

Introduction Section

Line 87: a reference about BCS is missing

Lines 96-99: In this last paragraph the authors should be more emphasize the interest in this CO system.

Methods

Line 108: The authors need to report the pharmacopoeia reference

Lines 118 – 125: In my opinion these equations should be reported at the beginning of section 2.3.

Line 149: The authors used as dissolution media 60% of ethanol, but this condition is not reported in Figure 6. The authors argue the use of this medium because the poorly aqueous solubility of these compounds (lines 436-439), but not always the results obtained in alcoholic media can be extrapolate to aqueous media. UV detector was used in the HPLC quantification, has been tested a MS detector? Moreover, the use of biorelevant media as FaSSIF or FeSSIF can improve the solubility of the drug and the dissolution test is closer to the in vivo conditions.

Line 175-176: Can the authors explain why they use 20 µL of SIM? Is added together with the plasma or is used in an independent solution as a control?

Line 199: add (HFD) after high-fat diet

Line 197: The mice in the other three groups. Change three by two

Lines 216-222: Is new this protocol? If yes, has it been validated? If not, a reference should be included.

Results

TGA analysis. Figure 1 only show the TGA for the crystalline compounds. Why haven't they done it for PM and/or CO mixtures? Do the authors think that TGA information is a relevant one or the information obtained with DSC is enough?

Lines 261-262: The two sentences are the same. Choose one of them.

Line 276: Fig 3A

Figure 3B: Add the number of 1728 and 1723 bands.

Line 280: is coamorphous or amorphous?

Lines 291-292: In my opinion this sentence should be introduce in line 282, when is the first time that the broad peak is appear

Lines 303-307 and Figure 5: A priori, the results showed in figure 5 do not evidence the better stability of CO in front the amorphous drugs. The authors should better demonstrate this affirmation. What means RH in the figure caption?

Figure 7 and Table 1: What means the error bars? Why are they in one direction? How are the PK parameters obtained? Which model are used and how about the goodness fit?

Figure 10 is not reported in the text

Discussion

The first part of the discussion is already explained in the introduction and/or results. It can be simplified.

Lines 444- 447: Can be remove. This information is in Result section and do not report any new information to discuss.

Author Response

Response to Review 2 Comments

One of my objections is that the manuscript is too long, and some aspects or comments are repeated in several sections. The authors should try to reduce the text without losing its integrity. The Abstract should be rewriting as the Instructions for Authors recommendations.

Answer: Thanks for the reviewer’s comment. We tried to make the manuscript concise and have revised the format of the abstract.

Some other aspects to be considered point by point:

The structures of the compounds should be reported

Answer:The chemical structures of Simvastatin (SIM) and Ezetimibe (EZE) have been reported in the Scheme 1.

Introduction Section

Line 87: a reference about BCS is missing

Answer: We have added the reference for BCS.

Lines 96-99: In this last paragraph the authors should be more emphasize the interest in this CO system.

Answer: We thank the reviewer for the comment. The paragraph was rewritten as: In clinical practice, ezetimibe (EZE) and simvastatin (SIM) are commonly used to decrease cholesterol, but their therapeutic effectiveness is significantly hampered by their poor water solubility and limited bioavailability. In this study, we created a CO system of SIM and EZE based on the different mechanisms of their cholesterol-lowering effects and the defects in the dissolution and absorption of the two drugs. The formation of CO can stabilize the high-energy binary amorphous mixture, make it easier for drug monomers to be absorbed, and reduce the levels of cholesterol in mouse plasma.

Line 108: The authors need to report the pharmacopoeia reference

Answer: we have added the reference: http://www.uspbpep.com/usp32/pub/data/v32270/usp32nf27s0_test-solutions.html

Lines 118 – 125: In my opinion these equations should be reported at the beginning of section 2.3.

Answer: Gordon–Taylor equation for theoretical Tg calculation is shifted to section 2.3.

Line 149: The authors used as dissolution media 60% of ethanol, but this condition is not reported in Figure 6. The authors argue the use of this medium because the poorly aqueous solubility of these compounds (lines 436-439), but not always the results obtained in alcoholic media can be extrapolate to aqueous media. UV detector was used in the HPLC quantification, has been tested a MS detector? Moreover, the use of biorelevant media as FaSSIF or FeSSIF can improve the solubility of the drug and the dissolution test is closer to the in vivo conditions.

Answer: All in vitro dissolution tests were performed in powder form. Both EZE and SIM belong to the BCS class II, and as a result, they have the propensity to float in an aqueous media. We agree with the reviewer that either FaSSIF or FeSSIF might potentially improve the solubility of the compound. Despite this, either FaSSIF or FeSSIF may be useful when the drugs are in a sink condition. We used 60% ethanol as a dissolution medium because EZE and SIM were in powder form. The dissolution condition has been added to the caption of Figure 6.

Line 175-176: Can the authors explain why they use 20 µL of SIM? Is added together with the plasma or is used in an independent solution as a control?

Answer: The 20 µL was the volume of the standard solution (5 mM). We estimated the possible concentrations of SIM in the blood, then prepared a stock solution and diluted it to a series of concentrations to prepare SIM’s plasma c~t standard curve.

The 20 µL standard solution was added with the plasma for subsequent standard curve drawing.

Line 199: add (HFD) after high-fat diet

Answer: We are thankful to the reviewer for this comment, and we corrected this problem.

Line 197: The mice in the other three groups. Change three by two

Answer: We are thankful to the reviewer for this comment, and this mistake has been corrected.

Lines 216-222: Is new this protocol? If yes, has it been validated? If not, a reference should be included.

Answer: We referred to a method reported in literature and made certain modifications of the method to conduct our experiments. The reference was added.

Sohn CW, et al. High temperature- and high pressure-processed garlic improves lipid profiles in rats fed high cholesterol diets. J Med Food. 2012 May;15(5):435-40.

Results

TGA analysis. Figure 1 only show the TGA for the crystalline compounds. Why haven't they done it for PM and/or CO mixtures? Do the authors think that TGA information is a relevant one or the information obtained with DSC is enough?

Answer: We are thankful to the reviewer for this comment. The CO mixture is prepared by the melt-quenched technique. So the intention of doing TGA experiments is to check whether the individual drugs are thermally stable or not. We added the explaination into the text: In order to test the thermal stability of drugs for preparing CO by the melt-quenched method, the TGA of crystalline SIM and EZE is shown in Figure 1.

Lines 261-262: The two sentences are the same. Choose one of them.

Answer: We are thankful to the reviewer for this comment, and we removed the second sentence (line 262) from the manuscript.

Line 276: Fig 3A

Answer: We are thankful to the reviewer for this comment, and we changed 2. A to 3. A.

Figure 3B: Add the number of 1728 and 1723 bands.

Answer: The numbers have been added into the Figure 3.

Line 280: is coamorphous or amorphous?

Answer: We thank the reviewer for pointing this out. It is amorphous, the correction has been made.

Lines 291-292: In my opinion this sentence should be introduce in line 282, when is the first time that the broad peak is appear

Answer: We are thankful to the reviewer for the suggestions. We moved the sentence to the place that the broad peak is appear for the first time.

Lines 303-307 and Figure 5: A priori, the results showed in figure 5 do not evidence the better stability of CO in front the amorphous drugs. The authors should better demonstrate this affirmation. What means RH in the figure caption?

Answer: We thank the reviewer for these comments. The CO samples are more stable than the amorphous samples of individual components at 40 ℃. For clarity and better comparison, we labelled the small crystalline peaks appeared in the amorphous SIM and EZE stored at 40 ℃ for 30 days. RH means room huminity.

Figure 7 and Table 1: What means the error bars? Why are they in one direction? How are the PK parameters obtained? Which model are used and how about the goodness fit?

Answer: The error bars were expressed by standard deviation (SD) here. To keep the graph clear to see, we used the same direction to present the SD, and it’s a plotting method being used by other researchers.

We used software DAS 2.0 to fit the pharmacokinetic data of mice.We used different fitting methods to fit different models by software DAS 2.0, and the results showed that the one-compartment model was most suitable and had the best fitting degree.

Figure 10 is not reported in the text

Answer: We are thankful to the reviewer for the comment, and we have reported the Figure 10 in the text.

Discussion

The first part of the discussion is already explained in the introduction and/or results. It can be simplified.

Answer:The discussion was simplified.

Lines 444- 447: Can be remove. This information is in Result section and do not report any new information to discuss.

Answer: We are thankful to the reviewer for the comment, and those two sentences were removed.

Round 2

Reviewer 1 Report

I am content with the changes made

Author Response

1: Abstract: Although the authors have made an effort to improve it, there are some aspects to be considered. Lines 39 – 45 contain very specific information that is not necessary in the abstract.  Line 45-48 can be unified with line 34 and 35.

Answer: Thanks for the reviewer’s comment. We have revised the abstract section of our manuscript.

2: This commend should be included in the text or in Table 1

 “We used software DAS 2.0 to fit the pharmacokinetic data of mice.We used different fitting methods to fit different models by software DAS 2.0, and the results showed that the one-compartment model was most suitable and had the best fitting degree”

Answer: Thanks for the reviewer’s comment. We have added information on the software and methods we used to line 331.

Reviewer 2 Report

Dear Editor,

The answers given by the authors are well argued and the changes done improve notably the manuscript. Nevertheless, there are still some few aspects to considerer before the publication of the manuscript:

1: Abstract: Although the authors have made an effort to improve it, there are some aspects to be considered. Lines 39 – 45 contain very specific information that is not necessary in the abstract.  Line 45-48 can be unified with line 34 and 35.

2. This commend should be included in the text or in Table 1

 “We used software DAS 2.0 to fit the pharmacokinetic data of mice.We used different fitting methods to fit different models by software DAS 2.0, and the results showed that the one-compartment model was most suitable and had the best fitting degree

Author Response

(The authors gave the same response as above.)
